# Association of Hyperkalemia and Hypokalemia with Patient Characteristics and Clinical Outcomes in Japanese Hemodialysis (HD) Patients

**DOI:** 10.3390/jcm12062115

**Published:** 2023-03-08

**Authors:** Masao Iwagami, Yuka Kanemura, Naru Morita, Toshitaka Yajima, Masafumi Fukagawa, Shuzo Kobayashi

**Affiliations:** 1Department of Health Services Research, Institute of Medicine, University of Tsukuba, Tsukuba 305-8575, Japan; 2Faculty of Epidemiology and Population Health, London School of Hygiene and Tropical Medicine, London WC1E 7HT, UK; 3Cardiovascular, Renal, and Metabolism, Medical Affairs, AstraZeneca K.K., Osaka 530-0011, Japan; 4Division of Nephrology, Endocrinology, and Metabolism, Tokai University School of Medicine, Isehara 259-1193, Japan; 5Kidney Disease and Transplant Center, Shonan Kamakura General Hospital, Kamakura 247-8533, Japan

**Keywords:** dyskalemia, maintenance hemodialysis, hypokalemia, hyperkalemia

## Abstract

This study aimed to examine the characteristics and clinical outcomes of Japanese hemodialysis patients with dyskalemia. A retrospective study was conducted using a large Japanese hospital group database. Outpatients undergoing thrice-a-week maintenance hemodialysis were stratified into hyperkalemia, hypokalemia, and normokalemia groups based on their pre-dialysis serum potassium (sK) levels during the three-month baseline period. Baseline characteristics of the three groups were described and compared for the following outcomes during follow-up: all-cause mortality, all-cause hospitalization, major adverse cardiovascular events (MACE), cardiac arrest, fatal arrythmia, and death related to arrhythmia. The study included 2846 eligible patients, of which 67% were men with a mean age of 65.65 (SD: 12.63) years. When compared with the normokalemia group (*n* = 1624, 57.06%), patients in the hypokalemia group (*n* = 313, 11.00%) were older and suffered from malnutrition, whereas patients in the hyperkalemia group (*n* = 909, 31.94%) had longer dialysis vintage. The hazard ratios for all-cause mortality and MACE in the hypokalemia group were 1.47 (95% confidence interval [CI], 1.13–1.92) and 1.48 (95% CI, 1.17–1.86), respectively, whereas that of death related to arrhythmia in the hyperkalemia group was 3.11 (95% CI, 1.03–9.33). Thus, dyskalemia in maintenance hemodialysis patients was associated with adverse outcomes, suggesting the importance of optimized sK levels.

## 1. Introduction

Despite the rapid progress of dialysis treatments and techniques, patients who are undergoing hemodialysis (HD) still have a poor prognosis. The mortality rate of maintenance dialysis patients in the United States continues to be unacceptably high and was reported at approximately 20% per year in 2020 [1,2,3]. On the other hand, according to data from the Japanese Society for Dialysis Therapy in 2020, the mortality rate for chronic dialysis patients in Japan is roughly half of that in the US at about 10%. This difference is difficult to explain solely by differences in dialysis methods and techniques, and moreover, even after considering the differences in socioeconomic factors and comorbidities, there are racial/ethnic differences in the mortality of maintenance dialysis patients [4,5,6,7,8,9].

Recent studies have shown that, compared with serum potassium (sK) levels of 4.6–4.9 mEq/L, sK levels > 5.6 mEq/L involve a higher risk of both all-cause mortality and cardiovascular mortality caused by arrhythmia in end-stage renal disease (ESRD) patients receiving HD therapy [10,11,12]. However, prior studies using large HD cohort data revealed that the distribution of sK levels is associated with mortality differently across race/ethnicity in maintenance HD patients [13,14]. According to these reports, higher sK levels at pre-dialysis were associated with higher mortality risk in Caucasian and African-American maintenance HD patients, whereas lower sK levels at pre-dialysis were associated with higher mortality risk in Hispanic patients. Furthermore, the Dialysis Outcomes and Practice Patterns Study (DOPPS) showed that the prevalence and severity of hyperkalemia varied by country [15]. In addition, hyperkalemia excursions over a 4-month period were associated with higher mortality risk in North America and Europe among maintenance HD patients; however, this was not the case in Japan [15]. Therefore, the relationship between the backgrounds and prognosis in patients with dyskalemia on maintenance HD remains to be elucidated in Japan.

To evaluate the impact of dyskalemia on the clinical prognosis of Japanese maintenance HD patients, we designed a comprehensive comparative cohort study, employing a large retrospective multicenter hospital-based database. The aim of this study is to examine the prevalence, incidence, demographics, treatment patterns, comorbidities, nutrition status, and clinical outcomes in Japanese HD patients with hyperkalemia and hypokalemia via comparison to those with normokalemia.

## 2. Materials and Methods

### 2.1. Study Design

This is a non-interventional, retrospective, cohort study using electronic health records derived from the Tokushu-kai information system of the Tokushu-kai group hospitals in Japan. Tokushu-kai group is one of the largest medical networks, comprised of 71 hospitals spread across Japan [16,17,18]. In this study, 63 hospitals among the 71 provided data sets. The overall data set included clinical records of diagnoses, diseases, treatment history, laboratory test results, and all medical procedures including surgery dates and types of examinations.

The study period was from 1 January 2010 to 31 March 2019. The first record of HD was defined as the date of the first HD treatment recorded in the Tokushu-kai information system, as recorded in the data source for each individual patient. The index date was defined as the date three months after the first record of HD for which data were available in the database. The baseline period was defined as the period of three months following the first record of HD for which data were available in the database. For the evaluation of comorbidities, the lookback period was defined as the period of up to 12 months before the first record of HD for which data were available in the database. The follow-up period was from the index date up to the end of the study period or when data from individual patients were no longer available in the claims data set—whichever came first. 

### 2.2. Study Population and Sample Size

All patients with medical records of at least three months and more than one recorded sK level were extracted. The predefined inclusion criteria for the study included male or female patients aged ≥18 years at the time of their first HD treatment, patients undergoing maintenance HD three times a week for at least three months, patients with their first HD record, either prevalent or incident HD patients, and patients for whom sK value(s) were available at least once during the baseline period. Patients with less than three dialytic sessions per week were excluded from this study. In addition, we excluded the inpatient group, defined as patients with at least one record of hospitalization during the baseline period because their potassium levels are unlikely to reflect their baseline status due to acute conditions.

The hyperkalemia group (Hyper-K) was defined as patients who had pre-HD sK ≥ 5.1 mmol/L once during the short inter-dialytic interval or pre-HD sK > 5.4 mmol/L once during the long inter-dialytic interval among those who had sK ≥ 5.1 mmol/L twice at any interdialytic interval during the baseline period. Among patients who did not meet the definition of the Hyper-K group, a hypokalemia group (Hypo-K) was defined as patients with sK levels < 3.5 mmol/L at the pre-dialysis stage. A normokalemia group (Normo-K) was defined as patients who met neither the Hyper-K nor the Hypo-K criteria. As shown in Figure 1, patients who met the hyperkalemia criteria were defined as Hyper-K (*n* = 909). Among the non-Hyper-K group, patients who met the hypokalemia criteria were stratified into Hypo-K (*n* = 313) and Normo-K (*n* = 1624). In this study, 20 patients satisfying the criteria of both hyperkalemia and hypokalemia were stratified into the Hyper-K group based on the pre-defined grouping method of this study. 

### 2.3. Prevalence of Dyskalemia

The prevalence of hyperkalemia or hypokalemia during the baseline period in patients undergoing maintenance HD was estimated. Hyperkalemia and hypokalemia were collected with the same criteria as the study population definition. For this analysis, patients (*n* = 20) who had both hyperkalemia and hypokalemia during the baseline period contributed to the calculation of the prevalence of hyperkalemia and hypokalemia, respectively. The prevalence of dyskalemia was stratified into five groups based on dialysis vintage at baseline; the groups were <1 year, 1≤ to <2 years, 2≤ to <5 years, 5≤ to <10 years, and ≥10 years.

### 2.4. Covariate and Outcome Measures

Patient demographics, clinical characteristics, treatment patterns, and laboratory data were measured during the baseline period and comorbidities were evaluated during the lookback period [Appendix A].

The clinical outcomes measured during the follow-up period were all-cause mortality, MACE, hospitalization (all-cause), cardiac arrest, fatal arrythmia defined as ventricular tachycardia, torsade de pointes or ventricular fibrillation, and death related to arrhythmia. Death related to arrhythmia was defined as death within 3 days of a fatal arrythmia [Appendix A].

### 2.5. Statistical Analyses

The baseline characteristics and treatment of patients were summarized. Differences in the prevalence of dyskalemia among dialysis vintage groups were statistically tested by the chi-square test. The Benjamini–Hochberg Procedure was used to correct for multiple comparisons.

The hazard ratios (HRs) of clinical outcomes were analyzed by the Cox proportional hazard model. Crude and adjusted HRs were calculated. For the adjusted model, the covariates used were age, gender, dialysis vintage, index year, body mass index (BMI), albumin, C-reactive protein (CRP), urea reduction ratio (URR), serum calcium, serum phosphorus (centralized by subtracting the mean), serum phosphorus squared (squared after centralized), hemoglobin, HD treatment time, Charlson Comorbidity Index (CCI), and history of the following comorbidities: heart failure, diabetes, cardiovascular comorbidities, myocardial infarction, other acute ischemic heart diseases, atherosclerotic heart disease of native coronary artery, chronic ischemic heart disease, stroke, unstable angina, and angina pectoris unspecified. For evaluation of clinical events between dyskalemia groups and the Normo-K group, log-rank test was used for all-cause mortality, MACE, and death related to arrhythmia, while the Fina and Gray test was used for hospitalization, cardiac arrest, and fatal arrhythmia. A *p*-value of <0.05 was considered statistically significant. All data analyses were performed using R version 3.51 (R Foundation, Vienna, Austria).

## 3. Results

### 3.1. Prevalence of Dyskalemia

In total, 5444 patient data sets undergoing HD three times a week during the baseline period were extracted from the database; 2846 outpatients were included in the subsequent analysis [Figure 1]. The mean age of the 2846 outpatients was 65.65 (SD:12.63) years, 67% were male, and the median dialysis vintage was 3.07 years (min 0.10, max 37.29) [Table 1].

The prevalence of dyskalemia stratified by dialysis vintage at baseline is shown in Figure 2. As described in Section 2.3, twenty patients who met both hyperkalemia and hypokalemia criteria during the baseline period contributed to the calculation of the prevalence of both hyperkalemia and hypokalemia, respectively. Therefore, 909 (31.94%) and 333 (11.70%) patients met criteria for hyperkalemia and hypokalemia, respectively. The prevalence of hyperkalemia significantly increased with increasing dialysis vintage (*p* < 0.001), whereas the prevalence of hypokalemia significantly decreased with increasing duration of dialysis (*p* < 0.001) [Figure 2].

### 3.2. Baseline Outpatient Characteristics

The baseline characteristics of patients by group are shown in Table 1. Patients in the Hypo-K group (69.37 years, SD: 12.68) tended to be older than in the other groups (approximately 65 years), whereas dialysis vintage in the Hyper-K group (median 4.27 years) tended to be longer than in the other groups (1.49 and 2.70 years in the Hypo-K and Normo-K groups, respectively). All patients in all groups were using dialysate with a potassium concentration of 2.0 mEq/L. As for nutritional status, the Hypo-K group tended to have lower serum total protein, serum albumin, geriatric nutrition risk index (GNRI), and normalized protein catabolic rate (nPCR) than the other groups. These values tended to be in the lower range in all groups when compared with clinical norms [19]. Additionally, the percentage of patients receiving nutritional guidance was higher in the Hypo-K group (64.54%) than in the other groups (approximately 53%). Among inflammatory markers, CRP tended to be higher in the Hypo-K group (median 0.20 mg/dL) than in the other groups (median 0.10 mg/dL and 0.11 mg/dL in the Hyper-K and Normo-K groups, respectively). Among the recorded comorbidities, the prevalence of dementia (12.14%) and sarcopenia (12.78%) in the Hypo-K group were higher than those of the Hyper-K group (1.98% and 6.16%, respectively) and the Normo-K groups (3.63% and 7.20%, respectively). Conversely, arrhythmias tended to be recorded more frequently among the patients in the Hyper-K group compared to the Hypo-K and Normo-K group patients [Table 1].

Among the therapeutic agents used, renin–angiotensin–aldosterone system inhibitors (ACEi/ARB/MRA) were used by 59.85%, 44.41%, and 48.71% of the patients in the Hyper-K, Hypo-K, and Normo-K groups, respectively. Laxatives were used by 31.79%, 53.99%, and 38.42% of patients in the Hyper-K, Hypo-K, and Normo-K groups, respectively. Potassium adsorbents were used by 21.23% of patients in the Hyper-K group, of which 16.39% was calcium polystyrene sulfonate (CPS) and 5.72% was sodium polystyrene sulfonate (SPS) [Table 1].

### 3.3. Clinical Outcomes

During the follow-up period, 947 (33.27%) of the 2846 outpatients died, and the mortality rate was 7.78 per 100 person-years [Appendix A]. Hypo-K was significantly associated with mortality when compared with Normo-K (HR, 1.47; 95% CI, 1.13 to 1.92) but not when compared with Hyper-K (HR, 0.92; 95% CI, 0.78 to 1.09). The associations of Hypo-K with MACE were also significant when compared with Normo-K (HR, 1.48; 95% CI, 1.17 to 1.86) but not when compared with Hyper-K (HR, 0.97; 95% CI, 0.84 to 1.12) [Figure 3 and Table 2]. On the other hand, Hyper-K was significantly associated with death related to arrhythmia (HR, 3.11; 95% CI, 1.03 to 9.33) [Figure 3 and Table 2]. 

## 4. Discussion

This study sought to understand the characteristics of HD patients with dyskalemia and the association of dyskalemia with clinical outcomes among Japanese patients undergoing maintenance HD. 

### 4.1. Characteristics of Tokushu-kai Hospital Group Database

This study used electronic medical record data between 2010 and 2019 from the Tokushu-kai hospital network. The average age of HD patients in this study population was 68.09 years and 65.65 years before and after excluding the inpatient group, respectively. According to the Current Status of Chronic Dialysis Therapy report by the Japan Society of Dialysis Therapy [20], the average age of chronic dialysis patients showed an upward trend from 66.21 years in 2010 to 69.09 years in 2019. Therefore, the study population was considered almost consistent with the real-world HD patients in Japan. 

On the other hand, in terms of mortality, overall mortality in this study through the follow-up period was 947 (33.27%) of 2846 participants, which is 7.78 per 100 person-years. This mortality rate is higher than the data from JDOPPS [21], which found that 562 (14%) of 3967 participants died during the follow-up period, and the overall mortality rate was 6.7 per 100 person-years. This difference may be explained by the characteristics of the database used in this study. The hospital group from which this database is derived actively accepts emergency patients, including those in more severe conditions who have been rejected or have not been able to be treated in other dialysis clinics, and this may have led to the higher mortality rate seen in this study. 

### 4.2. Association of Hypo-K with Baseline Characteristics and Outcomes

In this study, 54% of the hypokalemia group used laxatives, suggesting an association between laxative use and hypokalemia. It has been reported that laxative use was not associated with risk of hypokalemia (K < 3.5 mEq/L) during the preceding 1-year pre-ESKD period. On the other hand, in a group of patients ≥65 years, the use of laxatives contributed to the higher risk of hypokalemia [22]. Despite this previous study, our study demonstrated that over half of the hypokalemia group used laxatives. Therefore, more attention should be paid to the sK trajectory when patients are treated with laxatives, particularly for older patients who may be more prone to laxative-induced hypokalemia than younger patients. 

Notably, malnutrition was more prevalent in patients with Hypo-K, in whom nutritional indicators such as serum albumin and total protein were below reference values (3.41 g/L and 6.36 g/dL, respectively, for Hypo-K patients versus reference values of >4.0 g/dL and 6.5 to 8.0 g/dL, respectively). Furthermore, the inflammation marker CRP was higher than the normal values, suggesting a tendency toward malnutrition and inflammation. Many HD patients suffer from protein–energy wasting, also known as uremic malnutrition, as defined by the International Society of Renal Nutrition and Metabolism [23,24,25]. In contrast to simple malnutrition such as starvation with low nutrient intake, malnutrition in dialysis patients results from increased catabolism due to the effects of inflammatory cytokines, which are more likely to complicate chronic inflammation and atherosclerotic disease and increase the risk of cardiovascular diseases and death [26,27]. The improvement of health-related quality of life (HR-QoL) is a priority issue in HD patients which require more detailed nutritional management, higher ADL, and better nursing care [28,29,30]. In fact, in the current study, the incidence of death and MACE was significantly higher in Hypo-K patients when compared with Normo-K patients, both with and without adjustment for potential confounding factors. 

A strong correlation between malnutrition/low ADL and poor life/health span prognosis in HD patients is also well known [31,32]. This study showed that Hypo-K patients were prone to malnutrition–inflammation status, suggesting that this status may influence clinical outcomes in Hypo-K patients in addition to the direct impact of hypokalemia. Dementia and sarcopenia were more common in the Hypo-K group, despite hypokalemia not necessarily directly affecting cognitive dysfunction; however, it was suggested that hypokalemia is associated with malnutrition, which can lead to sarcopenia and subsequent frailty, which ultimately can result in cognitive decline. Moreover, the consideration of nutritional disorders in maintenance HD patients is particularly important, as such disorders greatly affect vital prognosis and HR-QoL. Thus, nutritional and care-based interventions from nurses and nutritionists as well as the expansion of medical teams and care will be important topics of discussion in the future for better treatment [33]. Despite the fact that sK levels were recorded at pre-dialysis (instead of post-dialysis), about 11% of outpatient HD patients were still categorized in the Hypo-K group in this cohort. This study revealed that hypokalemia reflects a pathological condition with a poor prognosis caused by malnutrition–inflammation. Therefore, optimization of sK levels for the purpose of proper nutritional management and the improvement of ADL may contribute to the improvement of health/lifespan of HD patients. Further research is warranted in this area.

### 4.3. Association of Hyper-K with Baseline Characteristics and Outcomes 

In this study, the prevalence of Hyper-K increased with longer dialysis vintage, culminating at a rate of 39.72% in patients with a dialysis vintage of 10 years or more. According to the United States Renal Data System 2001 report on the number of patients undergoing HD in Japan and the United States by duration of HD, most patients in the United States had a dialysis vintage of two years or less, while many patients had a dialysis vintage of three years or more in Japan. Similarly, the number of patients on dialysis for 10 years or more was 0.17% in the United States and 24.1% in Japan [34]. In addition, according to the statistics of the Japan Society of Dialysis Therapy, 27.8% of patients have been on dialysis for more than 10 years. Furthermore, the proportion of patients with a dialysis vintage of 20 years or more was less than 1% in 1992 but had increased to 8.3% by 2017. These results suggest that the number of patients undergoing dialysis treatment for a longer period will increase in the future in Japan. Taken together with our results, further improvement will be needed for the management of hyperkalemia in Japanese patients on maintenance HD. 

In the Hyper-K group, the incidence of mortality and MACE was similar to that of the Normo-K group [15]. According to the data from the DOPPS study, the HR of all-cause mortality in hyperkalemia excursions with sK > 6.0 mEq/L over a 4-month period in Japan was 1.04 (95% CI, 0.78 to 1.39), whereas those in North America and Europe were 1.35 (95% CI, 1.23 to 1.48) and 1.44 (95% CI, 1.23 to 1.68), respectively [15]. Previously, Kim et al. reported racial and ethnic differences in mortality associated with sK levels; however, this study did not include Asian maintenance HD patients [29]. Therefore, it was hypothesized that the Asian HD population, especially the Japanese population, appears to better tolerate higher sK levels than the North American and European HD populations. While the underlying mechanism of racial and ethnic differences in sK levels remains unclear, this may be explained by differences in diet across regions and countries. Further studies are needed to determine the underlying mechanisms for the varying associations between sK level and mortality across race and ethnicity.

On the other hand, the incidence of death related to arrhythmia, defined as death within three days from fatal arrythmia, significantly increased in the Hyper-K group compared with the Normo-K group. According to the Current Status of Chronic Dialysis Therapy report by the Japan Society of Dialysis Therapy [20], potassium-poisoning/sudden death was responsible for 1.7% of the causes of death. It is known that the incidence of sudden cardiac death in Japanese patients is lower than that in the European population [35,36]. This is likely due to the lower complication rate of cardiovascular diseases such as coronary artery disease, congestive heart failure, and left ventricular hypertrophy in the Japanese population [37].

Chronic hyperkalemia not only causes fatal arrhythmia but also restricts the intake of fruits and vegetables, including those that are rich in potassium. Furthermore, fruits and vegetables also offer an abundance of other nutrients such as fiber, minerals, and short-chain fatty acids. These nutrients are associated with a lower risk of cardiovascular disease and mortality [38,39]. It has been reported that potassium intake correlates with the intake of necessary nutrients such as protein, fiber, and energy [40]. Therefore, if excessive nutritional restrictions are imposed as a treatment for hyperkalemia, it may cause malnourishment in HD patients, leading to worse outcomes. Consequently, the importance of properly treating hyperkalemia should be understood and disseminated.

For potassium management, dialysate with a potassium concentration of 2.0 mEq/L is commonly used in Japan, as was the case for all patients in this study and those reported in the DOPPS study [41]. It has been reported that a higher sK gradient is independently associated with a greater risk of all-cause hospitalizations and emergency department visits but not mortality, potentially due to a low number of events [42]. Therefore, to minimize sK fluctuations between pre- and post-dialysis, a personalized management of the sK level and the potassium concentration of the dialysate may be prudent for optimal dialysis treatments. The use of potassium adsorbent was 20% in the Hyper-K group in this study, despite a mean sK level of 5.44 mEq/L. According to data from the DOPPS study in Japan, potassium adsorbent was used in just 1.9% of cases of hyperkalemia, and the frequency was about 6% even when the sK level was >6.0 mEq/L continuing for 4 months or more [15]. In the REVEAL-HK study, which investigated the real-world condition of patients with hyperkalemia in Japan, the frequency of prescribing potassium adsorbent due to hyperkalemia was 37.6% in patients with CKD during the study period, further suggesting that chronic hyperkalemia may persist in many patients without active interventions such as potassium absorbents [43]. 

### 4.4. Limitations

This study has limitations, which are inherent due to its retrospective design, and is subject to several biases such as selection bias and confounding factors, despite adjustment. The data used in this study were limited to the Tokushu-kai group of hospitals. Therefore, there may be Tokushu-kai group-specific prescription and treatment patterns. In addition, in hospital-based databases in general, data for patients sent to other hospitals for emergency conditions, as well as those dying at home, cannot be captured. Therefore, absolute risk (differences) of the studied outcomes might be underestimated, whereas the relative risks between the groups (i.e., Hyper-K, Hypo-K, and Normo-K groups) are expected to be estimated correctly. Moreover, since we conducted the study based on the hypothesis that baseline sK impacts the subsequent long-term outcomes, we were unable to examine whether and to what extent change in sK status during the follow-up could affect the outcome. This study was also limited to the Japanese HD population and should be interpreted with caution for other countries, despite the rigorous definition of dyskalemia in the current study.

## 5. Conclusions

Despite standardized rigorous thrice-weekly dialysis therapy, dyskalemia was prevalent in the patients in this cohort. Hypo-K was characterized by older patients suffering from malnutrition, a higher incidence of all-cause mortality, and MACE. Hyper-K was characterized by a longer history of dialysis and a higher incidence of death related to fatal arrhythmia. Dyskalemia was associated with worse clinical outcomes compared with Normo-K. Therefore, this study emphasized the importance of controlling sK levels in HD patients while also maintaining the nutritional health of each patient. Eliminating patient barriers to better nutritional diets, in combination with the use of potassium binders for hyperkalemia, is expected to improve the worse clinical outcomes and frailty associated with poor nutritional status in maintenance HD patients.

## Figures and Tables

**Figure 1 jcm-12-02115-f001:**
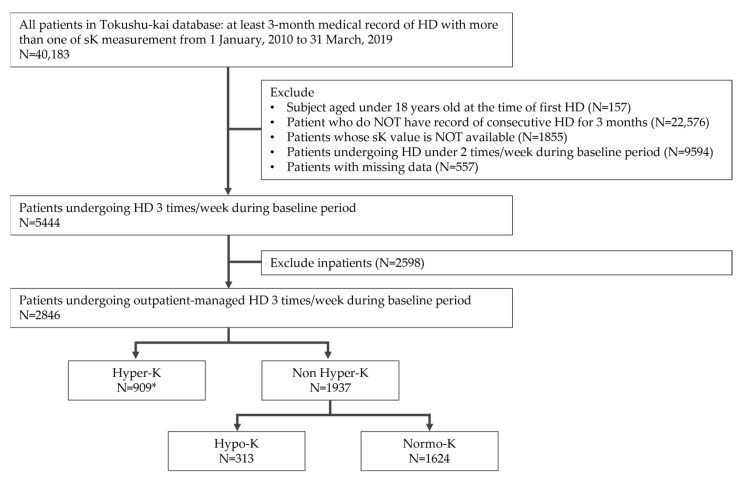
Flow diagram of patient inclusion in the study. Abbreviations: HD—hemodialysis; sK—serum potassium * Hyper-K group includes patients with both hyperkalemia and hypokalemia (*n* = 20).

**Figure 2 jcm-12-02115-f002:**
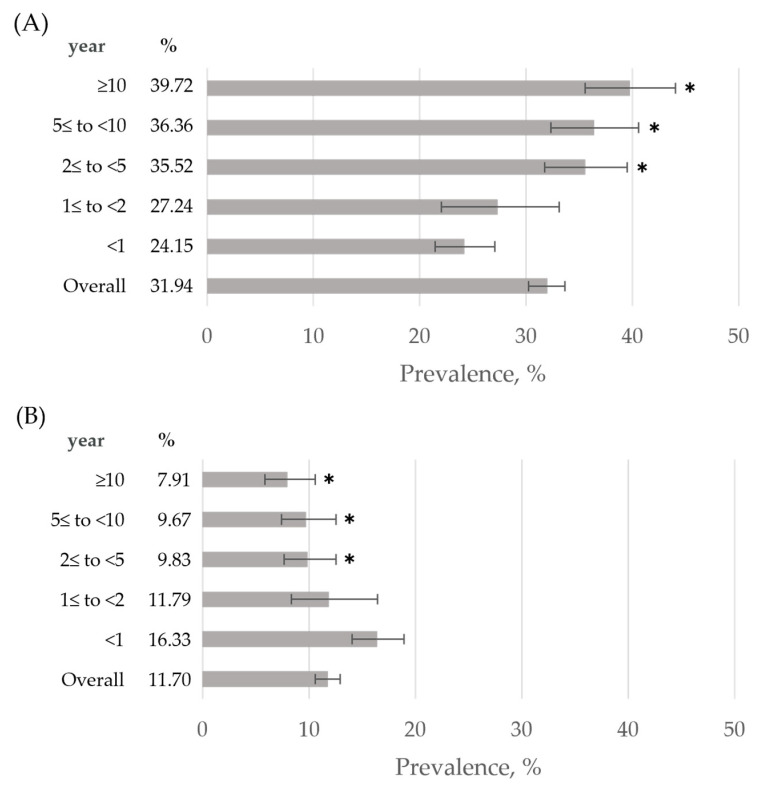
Prevalence of dyskalemia stratified by dialysis vintage at baseline period. (**A**) Prevalence of hyperkalemia (*n* = 909) during the baseline period by length of dialysis vintage. (**B**) Prevalence of hypokalemia (*n* = 333) during the baseline period by length of dialysis vintage [Hypo-K group (*n* = 313) and patients with both hyperkalemia and hypokalemia (*n* = 20)]. Error bars denote 95% confidence intervals. * *p* < 0.01 versus dialysis vintage of <1 year (the Benjamini–Hochberg Procedure was used to correct for multiple comparisons).

**Figure 3 jcm-12-02115-f003:**
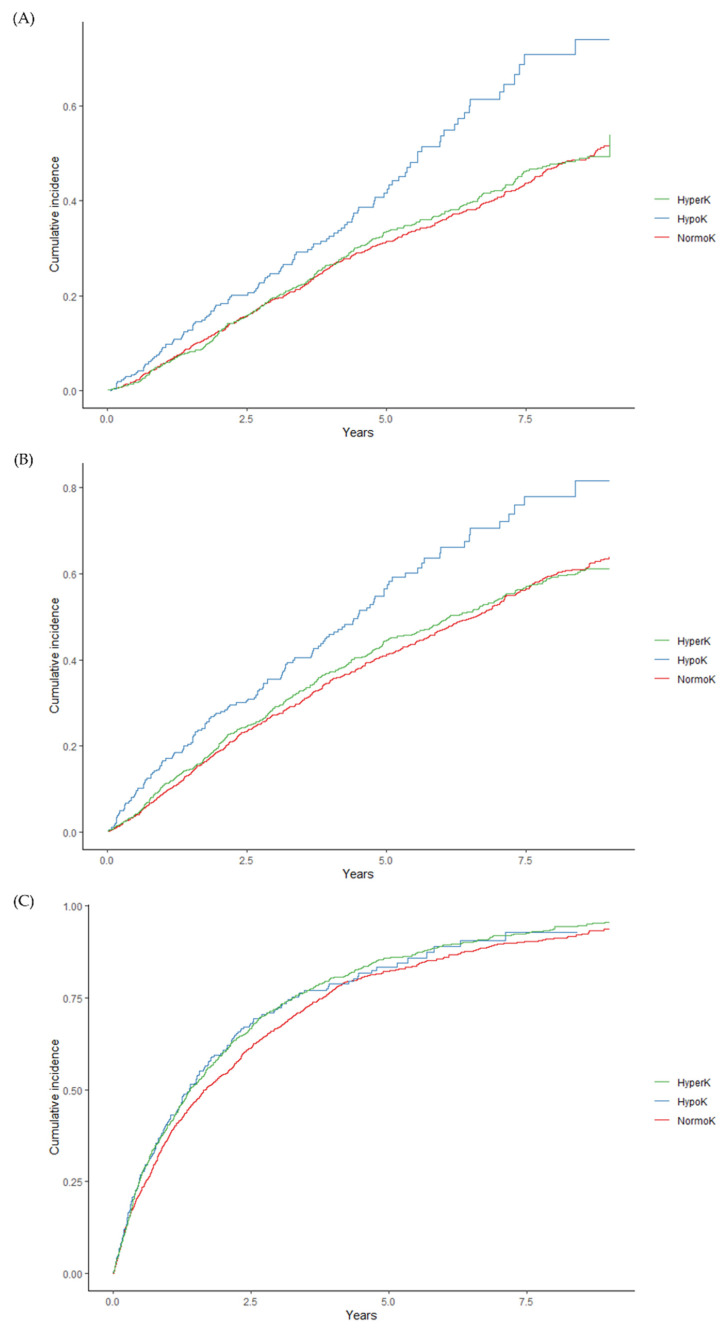
Cumulative incidence of clinical outcomes in Hyper-K, Hypo-K, and Normo-K. (**A**) All-cause mortality in Hyper-K (*p* = 0.900), Hypo-K (*p* < 0.001), and Normo-K; (**B**) MACE in Hyper-K (*p* = 0.700), Hypo-K (*p* < 0.001), and Normo-K; (**C**) hospitalization in Hyper-K (*p* = 0.018), Hypo-K (*p* = 0.381), and Normo-K; (**D**) cardiac arrest in Hyper-K (*p* = 0.201), Hypo-K (*p* = 0.364), and Normo-K; (**E**) fatal arrhythmia in Hyper-K (*p* = 0.827), Hypo-K (*p* = 0.214), and Normo-K; (**F**) death related to arrhythmia in Hyper-K (*p* = 0.040), Hypo-K (*p* = 1.000), and Normo-K. As evaluation of p-values, log-rank testing was used for all-cause mortality, MACE, and death related to arrhythmia, whereas the Fine and Gray test was used for hospitalization, cardiac arrest, and fatal arrhythmia.

**Table 1 jcm-12-02115-t001:** Baseline outpatient characteristics.

Characteristics	Overall	Hyper-K Group	Hypo-K Group	Normo-K Group
	*n* = 2846	*n* = 909 (31.94%)	*n* = 313 (11.00%)	*n* = 1624 (57.06%)
Age, year (mean ± SD)	65.65 ± 12.63	65.03 ± 11.95	69.37 ± 12.68	65.28 ± 12.87
Male, n (%)	1909 (67.08)	608 (66.89)	194 (61.98)	1107 (68.17)
BMI, kg/m^2^ (mean ± SD)	22.4 ± 8.79	22.25 ± 6.68	21.37 ± 3.59	22.65 ± 10.26
Dialysis vintage at baseline, median years (min, max)	3.07 (0.10, 37.29)	4.27 (0.23, 37.29)	1.49 (0.24, 27.09)	2.70 (0.10, 36.94)
2.0 mEq/L potassium dialysate, n (%)	2846 (100.00)	909 (100.00)	313 (100.00)	1624 (100.00)
Kt/V (mean ± SD)	1.37 ± 0.31	1.40 ± 0.30	1.33 ± 0.32	1.36 ± 0.31
Potassium, mEq/L (mean ± SD)	4.75 ± 0.77	5.44 ± 0.58	3.75 ± 0.65	4.55 ± 0.51
Calcium, mg/dL (mean ± SD)	8.73 ± 0.81	8.80 ± 0.81	8.51 ± 0.75	8.73 ± 0.82
Phosphorus, mg/dL (mean ± SD)	5.45 ± 1.48	5.85 ± 1.48	4.70 ± 1.55	5.36 ± 1.40
Hemoglobin, g/dL (mean ± SD)	10.79 ± 1.34	10.82 ± 1.30	10.64 ± 1.55	10.80 ± 1.32
Total protein, g/dL (mean ± SD)	6.48 ± 0.58	6.48 ± 0.57	6.36 ± 0.69	6.49 ± 0.56
Albumin, g/dL (mean ± SD)	3.63 ± 0.43	3.70 ± 0.39	3.41 ± 0.53	3.64 ± 0.42
Creatinine, mg/dL (mean ± SD)	9.77 ± 2.88	10.54 ± 2.73	7.69 ± 2.58	9.74 ± 2.82
URR, % (mean ± SD)	66.12 ± 8.34	66.57 ± 8.00	66.36 ± 8.83	65.81 ± 8.42
nPCR, g/kg/day (mean ± SD)	0.83 ± 0.24	0.84 ± 0.36	0.79 ± 0.07	0.83 ± 0.15
GNRI (mean ± SD)	93.08 ± 7.41	94.15 ± 7.26	90.51 ± 9.16	92.92 ± 7.07
Ferritin, ng/mL (median; min, max)	77.80 (0.07, 1970.0)	75.20 (0.19, 1900.9)	82.95 (0.07, 1970.0)	77.15 (0.17, 1961.3)
CRP, mg/dL (median; min, max)	0.11 (0.00, 39.47)	0.10 (0.01, 39.47)	0.20 (0.01, 11.03)	0.11 (0.00, 17.94)
Comorbidities, n (%)				
Diabetes	1558 (54.74)	484 (53.25)	191 (61.02)	883 (54.37)
Hypertension	2587 (90.90)	832 (91.53)	281 (89.78)	1474 (90.76)
Heart failure	1163 (40.86)	371 (40.81)	126 (40.26)	666 (41.01)
Cardiac arrest	6 (0.21)	5 (0.55)	1 (0.32)	0 (0.00)
Myocardial infarction	61 (2.41)	23 (2.53)	5 (1.60)	33 (2.03)
Stroke	383 (13.46)	123 (13.53)	34 (10.86)	226 (13.92)
Peripheral vascular diseases	974 (34.22)	331 (36.41)	94 (30.03)	549 (33.81)
Cerebrovascular diseases	797 (28.00)	249 (27.39)	93 (29.71)	455 (28.02)
Dementia	115 (4.04)	18 (1.98)	38 (12.14)	59 (3.63)
Sarcopenia	213 (7.48)	56 (6.16)	40 (12.78)	117 (7.20)
Medications, n (%)				
β-blockers	671 (23.58)	227 (24.97)	67 (21.41)	377 (23.21)
RAASi (ACEi/ARB/MRA)	1474 (51.79)	544 (59.85)	139 (44.41)	791 (48.71)
ACEi	168 (5.90)	69 (7.59)	22 (7.03)	77 (4.74)
ARB	1406 (49.40)	522 (57.43)	128 (40.89)	756 (46.55)
MRA	22 (0.77)	2 (0.22)	3 (0.96)	17 (1.05)
Laxative agent	1082 (38.02)	289 (31.79)	169 (53.99)	624 (38.42)
Potassium adsorbents (SPS/CPS)	384 (13.49)	193 (21.23)	20 (6.39)	171 (10.53)
CPS	311 (10.93)	149 (16.39)	14 (4.47)	148 (9.11)
SPS	91 (3.20)	52 (5.72)	7 (2.24)	32 (1.97)
Potassium supplements	16 (0.56)	1 (0.11)	7 (2.24)	8 (0.49)
Nutritional guidance	1555 (54.64)	489 (53.80)	202 (64.54)	864 (53.20)

Abbreviations: BMI—body mass index; URR—urea reduction ratio; nPCR—normalized protein catabolism rate; GNRI—geriatric nutrition risk index; CRP—C-reactive protein; RAASi—renin–angiotensin–aldosterone system inhibitor; ACEi—angiotensin-converting enzyme inhibitor; ARB—angiotensin receptor blocker; MRA—mineralocorticoid receptor antagonist; SPS—sodium polystyrene sulfonate; CPS—calcium polystyrene sulfonate.

**Table 2 jcm-12-02115-t002:** Association between serum potassium status and clinical outcomes in outpatients.

	N	100 Person-Year	Hazard Ratio (95% C.I.)	
			Crude	Adjusted ^a^
**All-cause mortality**				
Hyper-K (*n* = 909)	314	7.50	1.01 (0.87, 1.16)	0.92 (0.78, 1.09)
Hypo-K (*n* = 313)	120	11.29	1.58 (1.30, 1.94)	1.47 (1.13, 1.92)
Normo-K (*n* = 1624)	513	7.41	1.00 (Reference)	1.00 (Reference)
**MACE**				
Hyper-K (*n* = 909)	415	10.88	1.02 (0.91, 1.16)	0.97 (0.84, 1.12)
Hypo-K (*n* = 313)	152	16.29	1.57 (1.31, 1.87)	1.48 (1.17, 1.86)
Normo-K (*n* = 1624)	671	10.63	1.00 (Reference)	1.00 (Reference)
**Hospitalization**				
Hyper-K (*n* = 909)	751	41.97	1.14 (1.04, 1.24)	1.09 (0.98, 1.22)
Hypo-K (*n* = 313)	227	43.54	1.13 (0.98, 1.31)	1.13 (0.94, 1.37)
Normo-K (*n* = 1624)	1216	37.00	1.00 (Reference)	1.00 (Reference)
**Cardiac arrest**				
Hyper-K (*n* = 909)	41	0.98	1.24 (0.83, 1.86)	1.30 (0.80, 2.12)
Hypo-K (*n* = 313)	10	0.94	1.31 (0.66, 2.60)	1.41 (0.58, 3.38)
Normo-K (*n* = 1624)	54	0.78	1.00 (Reference)	1.00 (Reference)
**Fatal arrythmia**				
Hyper-K (*n* = 909)	32	0.78	1.25 (0.79, 1.97)	1.44 (0.86, 2.39)
Hypo-K (*n* = 313)	2	0.19	0.31 (0.08, 1.30)	0.28 (0.04, 2.16)
Normo-K (*n* = 1624)	43	0.63	1.00 (Reference)	1.00 (Reference)
**Death related to arrythmia**				
Hyper-K (*n* = 909)	11	0.26	2.65 (1.03, 6.84)	3.11 (1.03, 9.33)
Hypo-K (*n* = 313)	1	0.09	0.98 (0.12, 8.07)	2.65 (0.31, 22.89)
Normo-K (*n* = 1624)	7	0.10	1.00 (Reference)	1.00 (Reference)

To estimate the hazard ratio, the Normo-K group was used as the reference level. ^a^ Adjustments: age, gender, dialysis vintage, index year, body mass index (BMI), albumin, C-reactive protein (CRP), urea reduction ratio (URR), serum calcium, serum phosphorus (centralized by subtracting mean), serum phosphorus squared (squared after centralized), hemoglobin, HD treatment time, Charlson Comorbidity Index (CCI), and history of comorbidities: heart failure, diabetes, cardiovascular comorbidities, myocardial infarction, other acute ischemic heart diseases, atherosclerotic heart disease of native coronary artery, chronic ischemic heart disease, stroke, unstable angina, angina pectoris unspecified, heart failure, and diabetes.

## Data Availability

Not applicable.

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
