# Peer review of "Association of Hyperkalemia and Hypokalemia with Patient Characteristics and Clinical Outcomes in Japanese Hemodialysis (HD) Patients"

_jcm, 2023, doi:10.3390/jcm12062115_

Round 1

Reviewer 1 Report

Thank you for reviewing this manuscript. It is essential that in HD patients should reduce cardiovascular mortality with available techniques and medicines. The authors investigated the role of potassium level and mortality with a retrospective cohort study. The results are interesting; only a few questions were raised:

1.      In the hyper-K group was one patient who took potassium supplements. Why? If the patient had elevated potassium, why use a potassium supplement?

2.      Cardiac arrest significantly in the hypo-K group, but fatal arrhythmia is more associated with hyper-K level. What is the reason?

3.      It also needs to be clarified when the physicians realize the dyskalemia; why not change the hemodialysis fluid potassium concentration?

Author Response

Comments from Reviewer #1

  1. In the hyper-K group was one patient who took potassium supplements. Why? If the patient had elevated potassium, why use a potassium supplement?

Response: We appreciate reviewer’s question on patient baseline characteristics. Since it is database research, it is difficult to investigate the direct reason for prescription of potassium supplement in Hyper-K group. However, one possibility is that the patient developed hyperkalemia due to the use of the potassium supplement.

Since this study is real-world research using electronic medical record, such inappropriate patients may be included.

  1. Cardiac arrest significantly in the hypo-K group, but fatal arrhythmia is more associated with hyper-K level. What is the reason?

Response: We appreciate reviewer’s comment on clinical outcomes. Since the incidence of clinical outcomes is calculated based on the ICD-10 code, detail information about individual patient conditions, outcome and its interaction are hard to obtain. Therefore, it is difficult to show the confirmatory answer for your point. However, we speculate that one of the reasons may be that cardiac arrests occur by various reasons, and not just by arrythmia.

  1. It also needs to be clarified when the physicians realize the dyskalemia; why not change the hemodialysis fluid potassium concentration?

Response: We appreciate the reviewer’s question on dialysis treatment. As shown in this study and DOPPS data [Am J Kidney Dis. 2017;69(2):266-277], all of the Japanese dialysis facilities use dialysate that contains 2.0 mEq/L potassium, and adjusting hemodialysis fluid by each patient condition is not performed in clinical practice in Japan. This is probably mainly due to practical reasons such as hospital policies that most dialysis facilities in Japan adopt a centralized supply system where dialysate is prepared in one place and supplied to multiple bedside consoles (dialysis monitors), but ideally, we agree with you that adjustments should be considered in the future.

Reviewer 2 Report

This research uses a large database of > 2,800 chronic hemodialysis (HD) patients to investigate retrospectively the effects of dyskalemia [abnormal serum potassium concentration (sK)] during the first 3 month of chronic HD on clinical outcomes (all-cause mortality, all-cause hospitalization, major adverse cardiovascular events, cardiac arrest, fatal arrhythmias, and death related to arrhythmias) over the subsequent course of chronic HD.  I appreciate the identification of the gaps in the literature, the large number of patients studied, and the clear writing style.  I have a number of problems with the data presentation and the study design.

STUDY DESIGN

The authors are using labs (maybe only a single abnormal sK) from the first 3 months (“baseline period”) of HD to assign sK groups (Hypo-K, Hyper-K, or Normo-K) to predict outcomes over the entire course of HD, for some patients as long as 10 years of follow-up.  Is it possible or likely that a single (or several) abnormal sK value from the baseline period has impact after such a long period of time?  Since the data base for these patients is available, it might be better to follow sK values over all months of HD (rather than the first 3 months) and correlate these values with clinical outcomes.

In Section 2.2 (“study population and sample size”), the authors define the dyskalemic groups depending on when during the week the blood is drawn, ie after a short interdialytic interval vs after the long interdialytic interval.  In all HD units that I am familiar with, monthly blood is drawn at mid-week, ie on Wednesday for the Mon-Wed-Fri patients and Thursday for the Tues-Thur-Sat patients.  Please comment.

It is unclear to me why the patients with both hyperkalemia and hypokalemia during the baseline period were grouped with the Hyper-K only patients.  It would be interesting to compare clinical outcomes of the hyper-K/Hypo-K patients with the Normo-K patients, since the large swings of sK (from high to low or low to high) may foster arrhythmias.  This may be difficult statistically, since there were only 20 patients in this hyper-K/Hypo-K patient group.

I am confused by dialysis vintage In Figure 2 and Table 1.  In Table 1, dialysis vintage (ie years on HD) is included as a baseline characteristic, but dialysis vintage is calculated long after the baseline period by determining how long a patient remains on chronic HD.  Similarly, in Figure 2, prevalence of dyskalemia is stratified by dialysis vintage, but these occur at different times: dyskalemia during the baseline period and dialysis vintage only determined after many years.  Are the authors implying that dyskalemia in the first 3 months of dialysis has an effect on how long patients remain on chronic HD?

DATA PRESENTATION.

In Table 1, I see no statistics comparing patient characteristics between the 3 sK groups.

Figure 3 is very important, but the nature of the 3 lines (dots, dashes, linear) are difficult to distinguish between the lines.  Please remake these, so that it is easy for the reader to determine which line is which group.

Author Response

Comments from Reviewer #2

  1. The authors are using labs (maybe only a single abnormal sK) from the first 3 months (“baseline period”) of HD to assign sK groups (Hypo-K, Hyper-K, or Normo-K) to predict outcomes over the entire course of HD, for some patients as long as 10 years of follow-up. Is it possible or likely that a single (or several) abnormal sK value from the baseline period has impact after such a long period of time?  Since the data base for these patients is available, it might be better to follow sK values over all months of HD (rather than the first 3 months) and correlate these values with clinical outcomes.

Response: We thank the reviewer for pointing out this important view for this study. We agree that it is possible that sK status may change during the follow-up, and may affect the outcome. However, in our study, we assumed that the impact of the first 3 months sK level used for grouping would continue during the follow-up.

We consider there are study limitations in any study design. As for the study design you kindly proposed, we think there are also several limitations. First, modeling the relationship between sK level and outcomes over time is mathematically impractical, provided that it is unknown which time point of sK level affects outcomes at certain time. Second, the sK level during follow-up period may be an intermediate factor of exposure factor and outcomes, and adjusting this may result in overadjustment.

Overall, we agree that our study design also have study limitations and added your point as study limitation in our manuscript.

Changes in the manuscript:

P13: 4.4. Limitations

This study has limitations, inherent due to its retrospective design and is subject to several biases such as selection bias and confounding factors, despite adjustment. The data used in this study were limited to the Tokushu-kai group of hospitals. Therefore, there may be Tokushu-kai group-specific prescription and treatment patterns. In addition, in hospital-based databases in general, patients sent to other hospitals for emergency conditions, as well as those dying at home, cannot be captured. Therefore, absolute risk (differences) of the studied outcomes might be underestimated, whereas the relative risks between the groups (i.e., Hyper-K, Hypo-K, and Normo-K groups) are expected to be estimated correctly. Also, since we conducted the study based on the hypothesis that baseline sK impact the subsequent long-term outcomes, we may have ignored the possibility of change in sK status during the follow-up may affect the outcome. This study was also limited to the Japanese HD population and should be interpreted with caution for other countries despite the rigorous definition of dyskalemia in the current study.

  1. In Section 2.2 (“study population and sample size”), the authors define the dyskalemic groups depending on when during the week the blood is drawn, ie after a short interdialytic interval vs after the long interdialytic interval. In all HD units that I am familiar with, monthly blood is drawn at mid-week, ie on Wednesday for the Mon-Wed-Fri patients and Thursday for the Tues-Thur-Sat patients.  Please comment.

Response: We appreciate reviewer’s comment. As you pointed out, of approximately 70 hospitals which collected the data for this study, most of the blood samples were corrected during mid-week. However, in this real-world database, we  also found many patients who were collected blood samples outside mid-week (Mondays and Tuesdays). Therefore, we set a definition for sK levels at each dialysis interval and tried to collect all cases of potassium abnormalities.

  1. It is unclear to me why the patients with both hyperkalemia and hypokalemia during the baseline period were grouped with the Hyper-K only patients. It would be interesting to compare clinical outcomes of the hyper-K/Hypo-K patients with the Normo-K patients, since the large swings of sK (from high to low or low to high) may foster arrhythmias.  This may be difficult statistically, since there were only 20 patients in this hyper-K/Hypo-K patient group.

Response: We thank the reviewer for pointing out this important view for patients with dyskalemia. Originally we were interested in hyperkalemia, and divided patients into hyperkalemia and others. While building the study, we noticed that in the other group, there are patients with hypokalemia, which has high impact on clinical outcome, and stratified them. Since there were only 20 patients with hyperkalemia and hypokalemia, we believe that this didn't affect the result. However, we also think that the group with both hyperkalemia and hypokalemia is important. Therefore, we calculated the incidence of clinical outcomes of patients with hyperkalemia and hypokalemia for your reference, although it could not be statistically considered in this study due to the insufficient sample size.

Below table shows the incidence of clinical outcomes of patient with hyperkalemia and hypokalemia.

Cumulative incidence of clinical outcomes in patients with hyperkalemia and hypokalemia

Clinical outcomes

hyperkalemia and hypokalemia

(n=20)

N

100 PY

All cause mortality

at 6 months

1

10.27

at 1 years

3

15.94

at 3 years

6

13.60

at 5 years

9

15.14

over entire follow-up

9

12.48

MACE

at 6 months

1

10.27

at 1 years

5

27.12

at 3 years

8

20.12

at 5 years

10

19.11

over entire follow-up

10

15.91

Hospitalization

at 6 months

6

77.61

at 1 years

11

83.90

at 3 years

14

63.52

at 5 years

16

64.44

over entire follow-up

16

55.26

Cardiac arrest

at 6 months

1

10.27

at 1 years

1

5.31

at 3 years

1

2.27

at 5 years

2

3.36

over entire follow-up

2

2.77

Fatal arrythmia

at 6 months

0

0.00

at 1 years

0

0.00

at 3 years

0

0.00

at 5 years

0

0.00

over entire follow-up

0

0.00

Death related to arrythmia

at 6 months

0

0.00

at 1 years

0

0.00

at 3 years

0

0.00

at 5 years

0

0.00

over entire follow-up

0

0.00

  1. I am confused by dialysis vintage In Figure 2 and Table 1. In Table 1, dialysis vintage (ie years on HD) is included as a baseline characteristic, but dialysis vintage is calculated long after the baseline period by determining how long a patient remains on chronic HD.  Similarly, in Figure 2, prevalence of dyskalemia is stratified by dialysis vintage, but these occur at different times: dyskalemia during the baseline period and dialysis vintage only determined after many years.  Are the authors implying that dyskalemia in the first 3 months of dialysis has an effect on how long patients remain on chronic HD?

Response: We appreciate reviewer’s comment. Throughout the manuscript (both in Figure 2 and Table 1), length of HD is defined at baseline, not the prediction of the future dialysis vintage. We have clarified this in our revised draft.

Changes in the manuscript:

P5: Figure2

Figure 2. Prevalence of dyskalemia stratified by dialysis vintage at baseline period. (A) Prevalence of hyperkalemia (n=909) during the baseline period by length of dialysis vintage. (B) Prevalence of hypokalemia (n=333) during the baseline period by length of dialysis vintage [Hypo-K group (n=313) and patients with both hyperkalemia and hypokalemia (n=20)]. Error bars denote 95% confidence intervals. *p < 0.01 versus dial-ysis vintage of < 1 year (the Benjamini-Hochberg Procedure was used to correct for multiple comparisons).

P6: Table1

Characteristics

Overall

Hyper-K group

Hypo-K group

Normo-K group

n=2,846

n=909 (31.94%)

n=313 (11.00%)

n=1,624 (57.06%)

Age, year (mean±SD)

65.65±12.63

65.03±11.95

69.37±12.68

65.28±12.87

Male, %

67%

67%

62%

68%

BMI, kg/m2 (mean±SD)

22.4±8.79

22.25±6.68

21.37±3.59

22.65±10.26

Dialysis vintage at baseline,

median years (min, max)

3.07

(0.10, 37.29)

4.27

(0.23, 37.29)

1.49

(0.24, 27.09)

2.70

(0.10, 36.94)

2.0mEq/L potassium dialysate, %

100%

100%

100%

100%

Kt/V (mean±SD)

1.37±0.31

1.40±0.30

1.33±0.32

1.36±0.31

Potassium, mEq/L (mean±SD)

4.75±0.77

5.44±0.58

3.75±0.65

4.55±0.51

Calcium, mg/dL (mean±SD)

8.73±0.81

8.80±0.81

8.51±0.75

8.73±0.82

Phosphorus,mg/dL (mean±SD)

5.45±1.48

5.85±1.48

4.70±1.55

5.36±1.40

Hemoglobin, g/dL (mean±SD)

10.79±1.34

10.82±1.30

10.64±1.55

10.80±1.32

Total protein,g/dL (mean±SD)

6.48±0.58

6.48±0.57

6.36±0.69

6.49±0.56

Albumin, g/dL (mean±SD)

3.63±0.43

3.70±0.39

3.41±0.53

3.64±0.42

Creatinine,mg/dL (mean±SD)

9.77±2.88

10.54±2.73

7.69±2.58

9.74±2.82

URR, % (mean±SD)

66.12±8.34

66.57±8.00

66.36±8.83

65.81±8.42

nPCR, g/kg/day (mean±SD)

0.83±0.24

0.84±0.36

0.79±0.07

0.83±0.15

GNRI (mean±SD)

93.08±7.41

94.15±7.26

90.51±9.16

92.92±7.07

Ferritin, ng/mL

(median; min, max)

77.80

(0.07, 1970.0)

75.20

(0.19, 1900.9)

82.95

(0.07, 1970.0)

77.15

(0.17, 1961.3)

CRP, mg/dL

(median; min, max)

0.11

(0.00, 39.47)

0.10

(0.01, 39.47)

0.20

(0.01, 11.03)

0.11

(0.00, 17.94)

Comorbidities, n (%)

Diabetes

1,558 (54.74)

484 (53.25)

191 (61.02)

883 (54.37)

Hypertension

2,587 (90.90)

832 (91.53)

281 (89.78)

1,474 (90.76)

Heart failure

1,163 (40.86)

371 (40.81)

126 (40.26)

666 (41.01)

Cardiac arrest

6 (0.21)

5 (0.55)

1 (0.32)

0 (0.00)

Myocardial Infarction

61 (2.41)

23 (2.53)

5 (1.60)

33 (2.03)

Stroke

383 (13.46)

123 (13.53)

34 (10.86)

226 (13.92)

Peripheral vascular diseases

974 (34.22)

331 (36.41)

94 (30.03)

549 (33.81)

Cerebrovascular diseases

797 (28.00)

249 (27.39)

93 (29.71)

455 (28.02)

Dementia

115 (4.04)

18 (1.98)

38 (12.14)

59 (3.63)

Sarcopenia

213 (7.48)

56 (6.16)

40 (12.78)

117 (7.20)

Medications, n (%)

β-blockers

671 (23.58)

227 (24.97)

67 (21.41)

377 (23.21)

RAASi (ACEi/ARB/MRA)

1,474 (51.79)

544 (59.85)

139 (44.41)

791 (48.71)

ACEi

168 (5.90)

69 (7.59)

22 (7.03)

77 (4.74)

ARB

1,406 (49.40)

522 (57.43)

128 (40.89)

756 (46.55)

MRA

22 (0.77)

2 (0.22)

3 (0.96)

17 (1.05)

Laxative agent

1,082 (38.02)

289 (31.79)

169 (53.99)

624 (38.42)

Potassium adsorbents (SPS/CPS)

384 (13.49)

193 (21.23)

20 (6.39)

171 (10.53)

CPS

311 (10.93)

149 (16.39)

14 (4.47)

148 (9.11)

SPS

91 (3.20)

52 (5.72)

7 (2.24)

32 (1.97)

Potassium supplements

16 (0.56)

1 (0.11)

7 (2.24)

8 (0.49)

Nutritional guidance

1,555 (54.64)

489 (53.80)

202 (64.54)

864 (53.20)

  1. In Table 1, I see no statistics comparing patient characteristics between the 3 sK groups.

Response: We appreciate reviewer’s point. Regarding to the patient characteristics, the purpose of this study is to analyze the data descriptively, and the statistical comparison of each group was not the scope of the study. Since it was not pre-specified in our statistical analysis plan, we consider it is not appropriate to show the post-hoc analysis estimating p-values at this point in time.

  1. Figure 3 is very important, but the nature of the 3 lines (dots, dashes, linear) are difficult to distinguish between the lines.  Please remake these, so that it is easy for the reader to determine which line is which group.

Response: We appreciate reviewer’s point. In the original figure, each group is color-coded using green lines for Hyper-K, blue lines for Hypo-K, and red lines for Normo-K. Cumulative incidence of clinical outcomes in Hyper-K, Hypo-K and Normo-K are shown in below figures. (A) all-cause mortality (B) MACE (C) hospitalization (D) cardiac arrest (E) fatal arrhythmia (F) death related to arrhythmia.

Round 2

Reviewer 2 Report

This research uses a large database of > 2,800 chronic hemodialysis (HD) patients to investigate retrospectively the ability of dyskalemia [abnormal serum potassium concentration (sK)] during a 3-month period of chronic HD to predict clinical outcomes (all-cause mortality, all-cause hospitalization, major adverse cardiovascular events, cardiac arrest, fatal arrhythmias, and death related to arrhythmias) over the subsequent course of chronic HD.  I suggested in my original review a number of important changes to the study design, but the authors have not incorporated my suggestions into the revised manuscript.

STUDY DESIGN

Influence of sK in a 3-month period on clinical outcomes over a prolonged follow-up period.  Despite what the authors responded to my initial critique, I find it difficult to believe that a few sK levels during the baseline period can have an impact on clinical outcomes during a prolonged follow-up period, without assessing the sK levels during the follow-up period, which can be readily done. 

Timing of blood retrieval during the HD week.  In my initial critique, I was concerned that the sK levels from the baseline period were obtained at mid-week in some patients and after the weekend in other patients.  This is important.  The authors should only include patients whose blood was obtained at mid-week. 

Vintage.  When I read the original version of the manuscript, I assumed that the baseline period was the very first 3 months of HD for all the patients.  However, the authors may mean that the baseline period was the first 3 months of HD for which data were available in the Tokushu-kai data system and that many patients were on HD in another system for months or years before the baseline period of this study.  If so, the vintage would mean the time on HD prior to the baseline period of this study rather than the time on HD after the baseline period of this study.  Am I understanding this correctly?   The 2nd paragraph of the Materials and Method is very confusing in this regard; please rewrite.

Comparison of baseline characteristics of Hyper-K vs Hypo-K.  Despite what the authors responded to my initial critique, ANOVA is needed to compare baseline characteristics between Normo-K, Hyper-K, and Hypo-K patients.  This is a traditional approach in clinic research.

Patients with both Hyper-K and Hypo-K during the baseline period.  The 20 patients with Hyper-K and Hypo-K during the baseline period should be excluded rather than grouped with the Hyper-K patients.

Author Response

Comments from Reviewer #2

  1. Influence of sK in a 3-month period on clinical outcomes over a prolonged follow-up period. Despite what the authors responded to my initial critique, I find it difficult to believe that a few sK levels during the baseline period can have an impact on clinical outcomes during a prolonged follow-up period, without assessing the sK levels during the follow-up period, which can be readily done. We hope that the current revised manuscript is suitable for publication in Journal of Clinical Medicine.

Response: We thank the reviewer for pointing out this important view for this study. Although we could follow your suggestion, this change will lead to substantial change in study design. In our study, we assumed that the impact of the first 3 months sK level used for grouping would continue during the follow-up. We believe that it should be permissible from the point of view of an intention to treat (ITT) analysis to divide the baseline sK level into three groups and compare the subsequent outcomes. Moreover, as we explained in our first revision, the strategy proposed by the reviewer may not be necessarily better than the current analysis: First, modeling the relationship between sK level and outcomes over time is mathematically impractical, because is not unknown which time point of sK level affects outcomes at certain time. Second, the sK level during follow-up period may be an intermediate factor of exposure factor and outcomes, and adjusting this may result in overadjustment.

Again, it is our understanding that you made an important point, and to the best of our capability at the moment, we have added your point in limitation in the manuscript.

Changes in the manuscript from the first submission:

P13: 4.4. Limitations

This study has limitations, inherent due to its retrospective design and is subject to several biases such as selection bias and confounding factors, despite adjustment. The data used in this study were limited to the Tokushu-kai group of hospitals. Therefore, there may be Tokushu-kai group-specific prescription and treatment patterns. In addition, in hospital-based databases in general, patients sent to other hospitals for emergency conditions, as well as those dying at home, cannot be captured. Therefore, absolute risk (differences) of the studied outcomes might be underestimated, whereas the relative risks between the groups (i.e., Hyper-K, Hypo-K, and Normo-K groups) are expected to be estimated correctly. Also, since we conducted the study based on the hypothesis that baseline sK impact the subsequent long-term outcomes, we were unable to examine whether and to what extent change in sK status during the follow-up could affect the outcome. This study was also limited to the Japanese HD population and should be interpreted with caution for other countries despite the rigorous definition of dyskalemia in the current study.

  1. Timing of blood retrieval during the HD week. In my initial critique, I was concerned that the sK levels from the baseline period were obtained at mid-week in some patients and after the weekend in other patients.  This is important.  The authors should only include patients whose blood was obtained at mid-week.

Response: We appreciate reviewer’s comment. However, this method is commonly used for clinical trials for hyperkalemia in hemodialysis patients (J Am Soc Nephrol. 2019;30:1723-1733). In addition, changing the inclusion criteria for a study means that the study design and all of its results must be reviewed and interpreted from the ground up. Thus, we would like to keep the current inclusion criteria as is for the purpose of revision of the current manuscript.

  1. When I read the original version of the manuscript, I assumed that the baseline period was the very first 3 months of HD for all the patients.  However, the authors may mean that the baseline period was the first 3 months of HD for which data were available in the Tokushu-kai data system and that many patients were on HD in another system for months or years before the baseline period of this study.  If so, the vintage would mean the time on HD prior to the baseline period of this study rather than the time on HD after the baseline period of this study.  Am I understanding this correctly?   The 2nd paragraph of the Materials and Method is very confusing in this regard; please rewrite.

Response: We appreciate reviewer’s comment. As the reviewer mentioned, the dialysis vintage means the time on HD at the baseline period of this study rather than the time on HD after the baseline period of this study. We have clarified the Materials and Method. 

Changes in the manuscript:

P2: 2.1. Study design

The study period was from January 1st, 2010, to March 31st, 2019. The first record of HD was defined as the date of the first HD treatment recorded in the Tokushu-kai information system, as recorded in the data source for each individual patient. The index date was defined as the date three months after the first record of HD for which data were available in the database. The baseline period was defined as the duration of three months following the first record of HD for which data were available in the database. For the evaluation of comorbidities, the lookback period was defined as the period of up to 12 months before the first record of HD for which data were available in the database. The follow-up period was from the index date up to the end of the study period or when data from individual patients were no longer available in the claims data set, whichever came first.

P3: 2.3. Prevalence of dyskalemia

The prevalence of hyperkalemia or hypokalemia during the baseline period in patients undergoing maintenance HD was estimated. Hyperkalemia and hypokalemia were collected with the same criteria as the study population definition. For this anal-ysis, patients (n=20) who had both hyperkalemia and hypokalemia during the baseline period contributed to the calculation of the prevalence of hyperkalemia and hypokalemia, respectively. The prevalence of dyskalemia was stratified into five groups based on dialysis vintage at baseline which were < 1 year, 1 ≤ to < 2 years, 2 ≤ to < 5 years, 5 ≤ to < 10 years, and ≥ 10 years.

  1. Comparison of baseline characteristics of Hyper-K vs Hypo-K. Despite what the authors responded to my initial critique, ANOVA is needed to compare baseline characteristics between Normo-K, Hyper-K, and Hypo-K patients.  This is a traditional approach in clinic research.
  2. Patients with both Hyper-K and Hypo-K during the baseline period. The 20 patients with Hyper-K and Hypo-K during the baseline period should be excluded rather than grouped with the Hyper-K patients.

Response: We appreciate reviewer’s comment. We still believe that these analyses are not mandatory, because our goal was to describe the characteristics of the patients and the results of suggested analyses would not affect our conclusion. In addition, we need a few month's grace to change the statistical analysis protocol and receive approval in the studied institution for the non-preplanned, post-hoc analysis. Thus, we would like to ask the editor’s discretion on whether these analyses are really needed.
